# One-step construction of circularized nanodiscs using SpyCatcher-SpyTag

Shanwen Zhang[1,2], Qian Ren[1,2], Scott J. Novick[1], Timothy S. Strutzenberg [1], Patrick R. Griffin [1✉] & Huan Bao [1✉]

Circularized nandiscs (cNDs) exhibit superb monodispersity and have the potential to transform functional and structural studies of membrane proteins. In particular, cNDs can stabilize large patches of lipid bilayers for the reconstitution of complex membrane biochemical reactions, enabling the capture of crucial intermediates involved in synaptic transmission and viral entry. However, previous methods for building cNDs require multiple steps and suffer from low yields. We herein introduce a simple, one-step approach to ease the construction of cNDs using the SpyCatcher-SpyTag technology. This approach increases the yield of cNDs by over 10-fold and is able to rapidly generates cNDs with diameters ranging from 11 to over 100 nm. We demonstrate the utility of these cNDs for mechanistic interrogations of vesicle fusion and protein-lipid interactions that are unattainable using small nanodiscs. Together, the remarkable performance of SpyCatcher-SpyTag in nanodisc circularization paves the way for the use of cNDs in membrane biochemistry and structural biology.

---

[1] Department of Molecular Medicine, The Scripps Research Institute, Jupiter, Florida, USA. [2]These authors contributed equally: Shanwen Zhang, Qian Ren.
✉email: pgriffin@scripps.edu; hbao@scripps.edu

Cell-to-cell communication is essential for homeostasis and is mediated by membrane proteins MPs) that translate multiple inputs into a specific function[1,2]. Given their enormous potential in steering cell responses, MPs have long been the focus for mechanistic dissections and therapeutics development (e.g., cancer immunotherapy)[3,4]. These membrane-embedded molecules have complex biochemical properties to interact with hydrophobic lipids in the membrane and hydrophilic ligands in the cytosolic and extracellular spaces. Thus, characterizations of MPs are challenging, since extracting them from the lipid bilayer often invariably disrupts their function[5]. To tackle this challenge, the Sligar laboratory developed an elegant solution using engineered membrane scaffold proteins (MSPs) to enclose membrane targets in nanoscale lipid bilayers[6–9]. This technology, known as nanodiscs, has emerged as a powerful tool for functional and structural dissections of MPs[1]. In particular, nanodiscs could vastly increase the stability and monodispersity of the protein of interest in the lipid environment for NMR and single-particle cryo-EM studies[5,10].

MSPs are amphipathic proteins composed of a set of α-helixes derived from apolipoprotein A-1 (ApoA1). Although ApoA1 could wrap and transport lipids in vivo, its biophysical properties are polydisperse. Through rational design and protein engineering[6,7], MSPs were organized in a way that one face is hydrophilic to keep the protein soluble, and the other face is hydrophobic to encircle lipids and MPs. Over years of relentless engineering efforts, it is now possible to produce an array of monodisperse nanodiscs ranging from 6–16 nm in diameters[8,10].

However, it has been difficult to generate >20 nm nanodiscs for the reconstitution of membrane protein complexes, due to the increased heterogeneity of large nanodiscs. To restrain this variation, circularized nanodiscs (cNDs) were created to tight the control of nanodisc diameters and homogeneities[11,12]. These cNDs were produced through the sortase-mediated conjugation between the N- and C-termini of MSPs. Using this approach, Nasr et al. produced well-defined 50 nm nanodiscs for single-particle EM studies of membrane proteins[11]. Despite the power to reconstitute crucial reactions involved in synaptic transmission and viral entry, the sophisticated procedures and low yields for building these cNDs limit their broad application in membrane biochemistry. Perhaps the hydrophobic region of large MSPs exacerbates protein aggregation, and thus these proteins were mostly insoluble and subjected to degradation in vivo. To address

these problems, previous studies have also attempted using split inteins to circularize MSPs in cells[13]. Nevertheless, purification of the circularized MSPs requires multiple chromatography steps and is thus time-consuming. Moreover, it is unclear if the split-intein-based approach is amenable to building large cNDs with diameters over 30 nm.

Recently, the Howarth group developed the SpyCatcher-SpyTag technology for protein ligation[14]. This new method exhibits nearly diffusion-limited kinetics and has enabled circularization of several proteins with enhanced stabilities[15–19]. Inspired by these studies, we herein simplify the production of cNDs using SpyCatcher-SpyTag to circularize MSPs in cells, thereby by passing the in vitro protein refolding, sortase-mediated ligation, and repurification steps as required in the previous study. Furthermore, this approach greatly enhances the solubility and yield of circularized large MSPs to the same level as traditional small MSPs. Thus, we could readily build cNDs with diameters from 11 to over 100 nm, which allows us to reconstitute protein-lipid interactions and membrane fusion intermediates that are unattainable using small nanodiscs. Together, these circularized MSPs will significantly promote the use of cNDs for biochemical and structural studies of membrane proteins.

## Results

**Circularization of MSPs using SpyCatcher-SpyTag.** cNDs have profoundly advanced our work in the reconstitution of vesicle fusion[20,21]. However, the production of cNDs in the previous study is time-consuming, as one needs to carry out protein refolding, perform sortase-mediated ligation, and re-purify the circularized MSPs. To facilitate the application of cNDs, we set out to develop an alternative approach for nanodisc circularization. On this front, we are enlightened by the recent progress in the development of the SpyCatcher-SpyTag technology[22–24], which could link the N- to C-termini of the protein of interest in cells. We posit that this conjugation method would increase the yield of cNDs, since the spontaneous in vivo circularization of MSP could improve the protein solubility and stability by protecting its hydrophobic face from aggregation and degradation.

To do so, we fused the SpyCatcher and SpyTag to the N- and C-termini of MSP1D1[6,7], termed spMSP1D1 (Fig. 1a, b). Using a bacterial expression system, spMSP1D1 was efficiently produced as a major cyclized monomer with less than 15% of larger oligomers. We

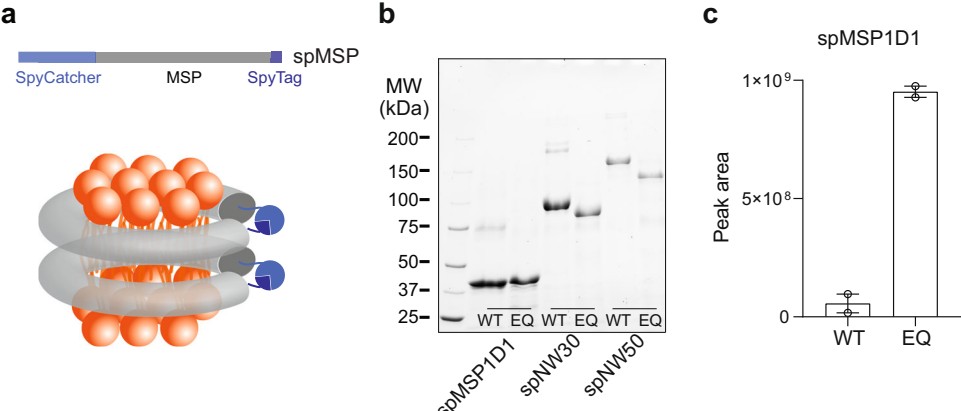

**Fig. 1 Circularization of MSPs via the SpyCatcher-SpyTag. a** The SpyCatcher (sapphire blue) and SpyTag (navy blue) were appended to the N- and C-termini of MSPs for spontaneous circularization of MSPs in cells (upper). Illustration of the circularized nanodiscs by the SpyCatcher-SpyTag (bottom). **b** Representative SDS-PAGE of circularized (WT) and non-circularized (EQ) MSPs. $n = 4$ independent experiments. **c** Quantification of the unreacted peptide in the WT and EQ mutant of spMSP1D1 using mass spectrometry. Data are shown as mean ± s.d., $n = 2$ independent experiments. Source Data are available as a source data file.

expressed the protein at 16 °C because more oligomers were formed at 37 °C (Supplementary Fig. 1a). This homogeneity could only be realized using low protein concentrations in the previous study via sortase-mediated protein ligation[11]. More importantly, spMSP1D1 exhibited a 2-fold higher yield than MSP1D1 (Supplementary Fig. 1b). To test if spMSP1D1 is circularized, we introduced two mutations that could abolish the covalent bond formation between SpyCatcher and SpyTag[22]. One mutant (EQ) is the substitution of Glu[77] to Gln in SpyCatcher, and the other (DA) is the substitution of Asp[117] to Ala in SpyTag. The results showed that spMSP1D1 is a bonafide circularized protein, as the non-circularized variants (EQ and DA) migrated at distinct positions on SDS-PAGE, and a mutant lacking the SpyTag was cleaved and degraded in cells (Fig. 1b and Supplementary Fig. 1c). Supporting these data, we further validated the isopeptide bond between the SpyCatcher and SpyTag using mass spectrometry (MS) (Fig. 1c). We identified peptides involved in the isopeptide bond formation, but we were unable to identify the linked peptide fragment. We believe this was the case because the linked tryptic peptides are large and highly charged due to missed cleavages (trypsin cannot cleave after an isopeptide bond). We used label-free quantitation to calculate that the circularization efficiency of spMSP1D1 was over 90%. This estimate is based on the change in the peak area of the unreacted peptide (Fig. 1c).

We then applied the same strategy onto large MSPs, known as NW30 and NW50[11], which could generate 15 and 50 nm nanodiscs via the sortase-mediated ligation approach, but suffers from low yields due to escalated aggregation and degradation elicited by their increased hydrophobic face. Using the SpyCatcher-SpyTag technology, we built spNW30 and spNW50. Gratifyingly, the yield of these two large circularized MSPs was improved by ~10 fold (Supplementary Fig. 1b), reaching the same level as MSP1D1. On SDS-PAGE, these two large MSPs, showed a molecular weight (MW) of 90 and 150 kDa, corresponding to the circularized monomer. Consistently, mutants defective in forming the isopeptide bond were again migrated at distinct positions (Fig. 1b). The amount of conjugated oligomers were less than 5%, as quantified on SDS-PAGE (Fig. 1b) and also confirmed using size-exclusion chromatography (SEC) (Fig. 2, top row). We noted that the sizes of the proteins on SEC were much bigger than SDS-PAGE. This is probably because the circularized MSPs have a hollow ring structure, so their MWs could not be accurately determined by SEC calibrated using globular protein markers. Together, we conclude that the SpyCatcher-SpyTag technology could enable the simple and straightforward circularization of homogenous MSPs with high efficiency.

**Nanodisc reconstitution and characterization**. Next, we tested if spMSP1D1, spNW30, spNW50 could form nanodiscs. To this end, we performed nanodisc reconstitutions using these MSPs and PC lipids. The reconstituted materials were separated by SEC using a Superose 6 column (Fig. 2), and fractions from the main peaks were analyzed by negative stain EM (Fig. 3) and a second SEC (Supplementary Fig. 2). Upon incorporation of optimal amounts of lipids, monodisperse protein-lipid particles formed with the three circularized MSPs (Figs. 2, 3 and Supplementary Fig. 2). Increasing or decreasing the lipid-protein ratios resulted in reduced yields of the main peaks and the formation of multiple species (Fig. 2), similar to previous studies of nanodisc assembly using non-circularized MSPs[6,7].

The MWs of these nanodiscs were estimated by SEC calibrated with a standard set of proteins (Fig. 2). Based on such calibration, the MWs of cNDs formed using spMP1D1, spNW30, spNW50 were 283, 717, and 1416 kDa, respectively. Assuming two copies of MSPs in each nanodisc, the estimated numbers of lipids per nanodisc were 259, 703, and 1486. To validate these calculated

values, we employed PC lipids dopped with 0.5% N-(lissamine rhodamine B sulfonyl)-1,2-dipalmitoyl-sn-glycero-3-phosphoetha-nolamine (rhodamine-PE). The results revealed a similar gradual increase of lipid: protein ratio in spMSP1D1, spNW30, and spNW50 reconstituted cNDs (Supplementary Fig. 3), indicating that large nanodiscs indeed encased more lipids.

We further characterized these nanodiscs using negative stain EM (Fig. 3a–c). Consistent with the SEC analyses, the particles formed by spMSP1D1, spNW30, spNW50 showed increased sizes and their average diameters were 11, 32, 53 nm, respectively. We are particularly amazed by the ability of spNW30 to produce 32 nm NDs, as NW30 would only generate 15 nm NDs using the sortase-mediated protein ligation approach[11]. This difference was not due to the altered assembly of spNW30 nanodiscs, because biochemical cross-linking and fluorescence fluctuation spectro-scopy experiments showed that spMSP1D1, spNW30, and spNW50 cNDs were all formed by two copies of MSPs (Supplementary Fig. 4). We suspected that the refolding process of NW30 in the previous study resulted in a much smaller size of cNDs as compared to our study.

**Extending the size range of cNDs**. The robust efficiency of the SpyCatcher-SpyTag in the circularization of MSPs prompted us to test if we could further use this approach to fine-tune the size range of cNDs. To do so, we simply truncated spNW30, thus producing spNW15 and spNW25 (Fig. 4a). In addition, we also attempted to make even larger MSPs by genetic fusion of NW30 and NW50 onto spNW50, yielding spNW80 and spNW100 (Fig. 4a). These four cyclized MSPs were readily produced with high yields and purity. We are again amazed by the ease of producing spNW80 and spNW100, as these two MSPs contain large hydrophobic domains and have an MW of 205 and 247 kDa, respectively. It is usually difficult to express proteins of these sizes in bacterial, yet the yields of spNW80 and spNW100 were on par with the much smaller 25 kDa MSP1D1.

Next, we tested the ability of spNW15, spNW25, spNW80, and spNW100 to build cNDs. spNW15 and spNW25 cNDs were purified by SEC (Supplementary Fig. 5), exhibiting the expected mean sizes of 15 and 24 nm as characterized by negative stain EM (Fig. 4b, c). However, cNDs reconstituted using NW80 and NW100 were beyond the exclusion limit of SEC, so they were isolated from aggregates and liposomes using co-flotation. The size distribution of spNW80 and spNW100 cNDs was much broader than smaller ones (Fig. 4d, e). This is probably due to the low-resolution separation by co-flotation, thus requiring the development of new approaches for the purification of large cNDs with diameters over 50 nm. Nevertheless, the diameters of spNW80 and spNW100 cNDs were clearly expanded beyond 80 and 100 nm, consistent with the predicted size of larger MSPs. Together, we concluded that the SpyCatcher-SpyTag circularized MSPs extended the range of cND sizes and could be used to build monodisperse large nanodiscs.

**Application of large cNDs**. The limited size of small nanodiscs precludes its application in the reconstitution of many membrane biochemical reactions. For example, using 6 and 13 nm nanodiscs, previous studies were able to isolate nascent fusion pores[25–27], a crucial intermediate state formed during vesicle exocytosis. In contrast, dilation of nascent fusion pores was prevented in these small nanodiscs, yet it is an important step to control cargo release in the secretory pathway. Liposome-based reconstitution approach could study pore dilation[28], but it is difficult to parse out nascent fusion pores from dilated ones. Recently, large cNDs formed using the sortase-mediated protein ligation approach allowed us to reconstitute pore dilation[20,21], however, only at the single-

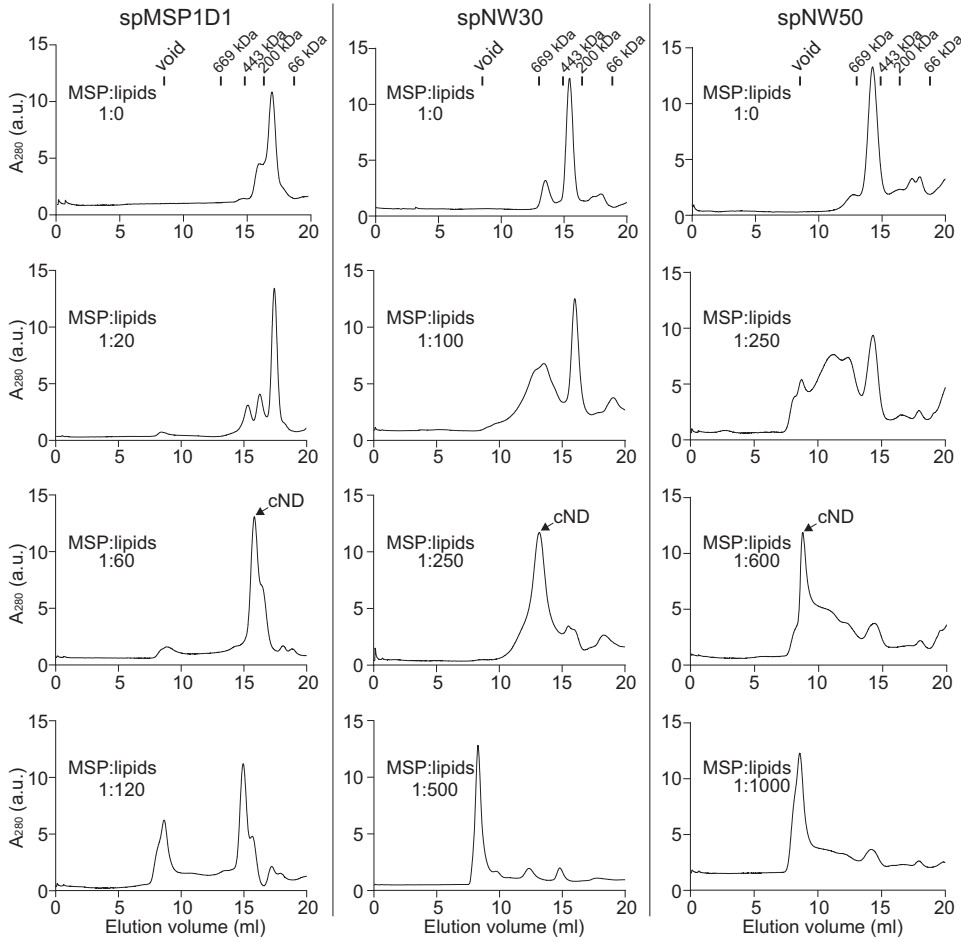

**Fig. 2 SEC of circularized MSPs and nanodiscs.** Circularized MSPs and reconstituted cNDs at the indicated protein: lipid ratios were fractioned using a Superose 6 10/300 column. From left to right: spMSP1D1, spNW30, spNW50. Fractions corresponding to NDs (arrowhead) were collected for further analysis. The proteins used for the calibration of the Superose 6 column are: thyroglobulin, 669 kDa; ferritin, 443 kDa; β-amylase, 200 kDa; bovine serum albumin, 66 kDa. a. u., arbitrary units. Source Data are available as a source data file.

molecule level, as the yield of these nanodiscs was too low for ensemble experiments. Here, we revisited this problem (Fig. 5a), since now we could prepare a higher amount of large cNDs with graded diameters from 11 to over 100 nm (Figs. 3 and 4).

Using nanodiscs bearing vesicle SNAREs (v-SNAREs) and liposomes harboring target membrane SNAREs (t-SNAREs), fusion pores could be formed, as revealed by the release of glutamate (Fig. 5b). In line with previous studies[29,30], the fusion pore was trapped in the initial open state using small spMSP1D1 nanodiscs, as a large cargo, 40 kDa fluorescein dextran, could not be released. In contrast, spNW25 and spNW50 encased nanodiscs harboring v-SNAREs could efficiently result in the release of fluorescein dextran (Fig. 5b). These reactions were dependent on the pairing of v- and t-SNAREs, as the release efficiencies of glutamate and fluorescein dextran were inhibited by over 20-fold upon addition of the cytoplasmic domain of v-SNAREs (cd-V) (Fig. 5b). Thus, pore dilation events were successfully captured using 24 and 53 nm cNDs in ensemble assays. Furthermore, we found that these cNDs exhibited much improved stability than liposomes or non-circularized nanodiscs for the reconstitution of membrane fusion (Supplementary Fig. 6). Since we could rapidly produce these large nanodiscs, structural studies of dilated fusion pores are now within reach using single-particle cryo-EM.

Moreover, we suspect that the expanded lipid bilayer of large nanodiscs could capture protein-lipid interactions that are not possible with small nanodiscs. In particular, the association of

proteins with lipids could result in high-order oligomers that surpass the capacity of the constrained surface area in small nanodiscs. To compare the utility of small and large nanodiscs for the characterization of protein-membrane interactions, we chose a set of bacterial and eukaryotic proteins: synaptagmin-1 (syt1), complexin-2 (cpx2), and glucose-specific enzyme IIA (EIIA$^{\text{Glu}}$), with lipid-binding affinities ranging from 10 nM to 100 μM[31–33]. Binding of these proteins to the lipids in nanodiscs was characterized using fluorescence spectroscopy (Fig. 6 and Supplementary Fig. 7). In comparison with small spMSP1D1 nanodiscs, the data showed that the apparent binding affinity of syt1 to large spNW50 nanodiscs was 3-fold higher and exhibited a hill coefficient of 2.1 (Fig. 6a–d and Supplementary Fig. 7a), suggesting that the cooperative binding and oligomer formation of syt1 occurred only on the expanded lipid surface[34–36]. These observations were not caused by the higher amount of lipids using large spNW50 nanodiscs, as plotting these data as a function of lipid concentrations yields similar results (Supplementary Fig. 7b). In addition, we were not able to detect the binding of cpx2 and EIIA$^{\text{Glu}}$ to lipids using small spMSP1D1 nanodiscs, whereas these two protein-lipid interactions could be readily captured using large spNW50 nanodiscs (Fig. 6e, f and Supplementary Fig. 7c, d). Together, these large cNDs, circularized using the SpyCatcher-SpyTag, are amenable to reconstituting complex membrane biochemical reactions that are unfeasible using small nanodiscs.

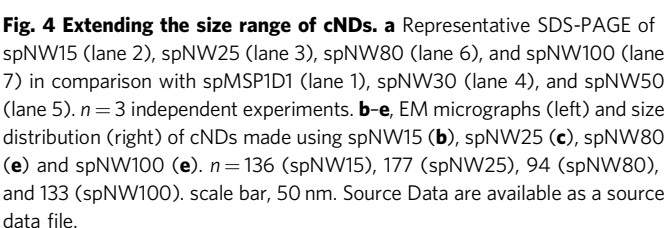

**Fig. 3 Characterization of purified cNDs by negative stain EM. a–c**, EM micrographs (left) and size distribution (right) of cNDs made using spMSP1D1 (**a**), spNW30 (**b**), and spNW50 (**c**). n = 117 (spMSP1D1 cNDs), 148 (spNW30 cNDs), 113 (spNW50 cNDs). scale bar, 50 nm. Source Data are available as a source data file.

## Discussion

In this manuscript, we presented a method to ease the construction of cNDs (Supplementary Fig. 8). Our approach could simplify the production of circularized MSPs into a single step, improve the yield of large cNDs by over 10 folds and extend the range of cND sizes from 11 to over 100 nm. This progress would profoundly promote the use of large cNDs and spark further engineering efforts to functionalize nanodiscs for a wide range of applications in the biochemical reconstitution of membrane biology.

Our approach removed the bottleneck of producing large cNDs. In previous studies, cNDs were developed to tight the control of nanodisc monodispersity and further increase its diameter to 50 nm. This is achieved by the sortase-mediated ligation of the N- to the C-termini of MSPs. The engineered proteins are mostly insoluble and non-circularized in cells, thereby requiring in vitro refolding and protein ligation for cyclization of large MSPs[11]. Although circularized MSPs have indeed improved the homogeneity of nanodiscs, building these cNDs was hampered by time-consuming multistep procedures and low yields. We suspect that the hydrophobic face of large MSPs tends to aggregate and also elicit degradation. To solve this problem, a previous study attempted to cyclize MSPs in cells via split inteins. This method could generate a sufficient amount of circularized MSPs for

**Fig. 4 Extending the size range of cNDs. a** Representative SDS-PAGE of spNW15 (lane 2), spNW25 (lane 3), spNW80 (lane 6), and spNW100 (lane 7) in comparison with spMSP1D1 (lane 1), spNW30 (lane 4), and spNW50 (lane 5). n = 3 independent experiments. **b–e**, EM micrographs (left) and size distribution (right) of cNDs made using spNW15 (**b**), spNW25 (**c**), spNW80 (**e**) and spNW100 (**e**). n = 136 (spNW15), 177 (spNW25), 94 (spNW80), and 133 (spNW100). scale bar, 50 nm. Source Data are available as a source data file.

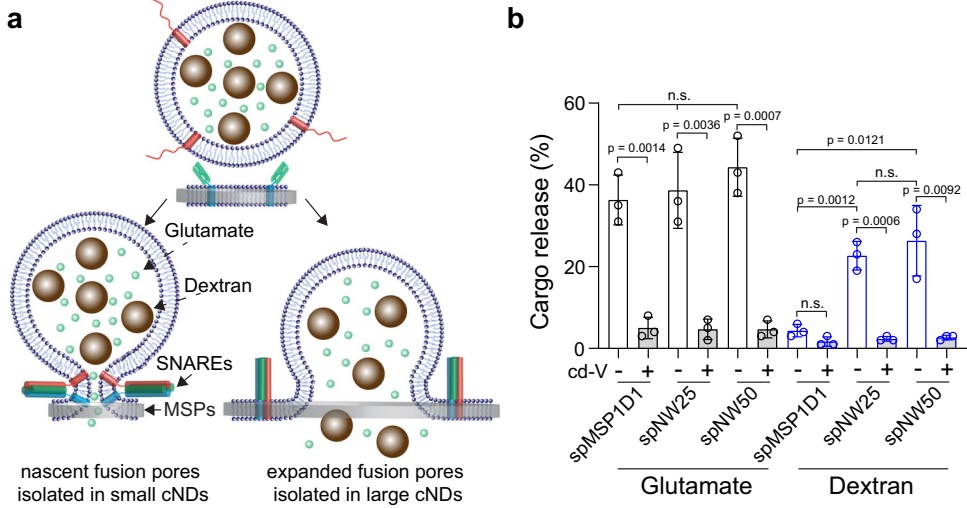

**Fig. 5 Application of cNDs in the reconstitution of fusion pores. a** Illustration of the reconstituted fusion pores in small and large nanodiscs. **b** Quantification of glutamate and fluorescent dextran release from fusion pores reconstituted using the indicated MSPs. Data are shown as mean ± s.d., $n = 3$ independent experiments. Statistics were determined using a two-tailed $t$-test. n.s., not significant. Source Data are available as a source data file.

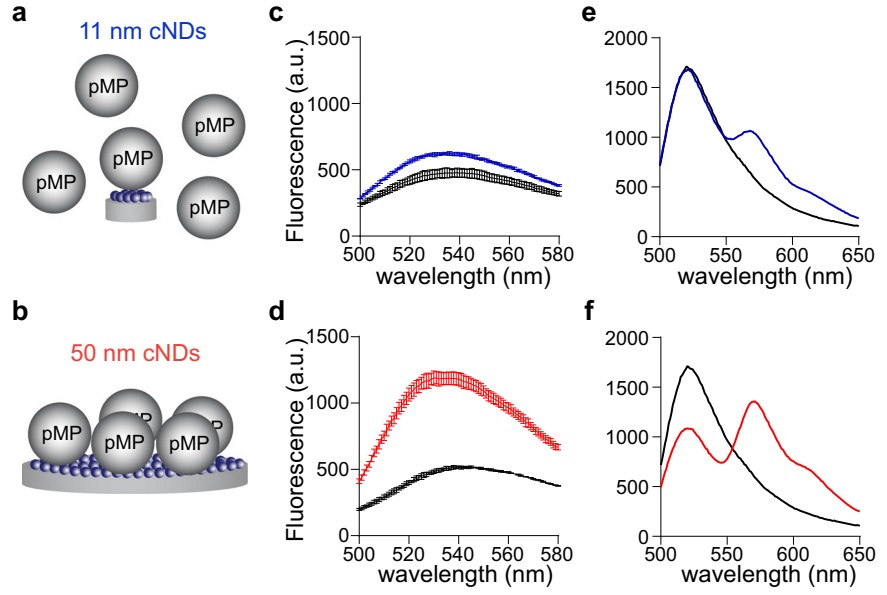

**Fig. 6 Application of cNDs for characterization of protein-lipid interactions. a–b**, Illustration of peripheral membrane protein (pMP) binding to small (**a**) and large (**b**) nanodiscs. **c–d**, Fluorescence emission spectrum of NBD-labeled syt1 (10 nM) in the absence (black) or presence of 11 (**c**, blue) and 50 (**d**, red) nm cNDs (1 µM). **e–f**, Fluorescence emission spectrum of OG-labeled EIIA^Glu (10 nM) in the absence (black) or presence of DiI-labeled 11 (**e**, blue) and 50 (**f**, red) nm cNDs (10 µM). Data are shown as mean ± s.d., $n = 3$ independent experiments. a. u., arbitrary units. Source Data are available as a source data file.

8–26 nm cNDs, but require several fractionation steps to isolate the protein[13]. Using SpyCatcher-SpyTag in our approach, MSP circularization spontaneously occurred in vivo and exhibited nearly 100% efficiency. As a consequence, spMSPs showed greatly improved solubility and yields (Fig. 1 and Supplementary Fig. 1), and only needed a single step for purification (Supplementary Fig. 8). This remarkable result exemplified the superior performance of the SpyCatch-SpyTag technology for protein ligation. With the continuous development of this powerful method[14], we expect that more sophisticated cNDs could be readily built for complex membrane systems.

Our work also demonstrated the remarkable plasticity of MSPs. After appending the SpyCather-SpyTag, all the circularized MSPs are still able to efficiently enclose lipids into nanodiscs (Figs. 2–4).

Our recent work was in line with this discovery; MSP fused with proximity labeling enzymes could still form nanodiscs and thus allowed us to probe membrane protein-lipid and protein-protein interactions via proximity labeling[37]. In addition, systematic truncation of MSP1D1 could yield stable and monodisperse small nanodiscs with diameters from 6–9 nm[10]. Furthermore, combining nanodiscs with DNA origami technology, a number of exciting lipid bilayer platforms were developed for the study of membrane biology[12,38,39]. Thus, we believe that an enormous unexplored potential to functionalize MSP remains to be realized.

Our work is consistent with previous studies showing that MSPs play a key role in determining the characteristics of nanodiscs[6,7,13]. We observed that the addition of SpyCatcher-SpyTag in nanodisc changed its physical and chemical properties.

The size and mass of spMSP1D1 nanodiscs (11 nm and 280 kDa) were higher than that of the MSP1D1 nanodiscs (9 nm and 200 kDa). In addition, the stability of cNDs was markedly enhanced as compared to non-circularized nanodiscs and liposomes (Supplementary Fig. 6). Therefore, the properties of nanodiscs could be fine-tuned by further engineering of MSPs.

The expanded lipid environment from large cNDs would enable characterizations of myriad membrane reactions that could not be characterized previously due to the size limitation of small nanodiscs. Here, we first showcased the power of circularized large nanodiscs in the reconstitution of fusion pore dilation (Fig. 5). We previously could only do so at the single-molecule level due to the low yields of the sortase-mediated protein ligation method. Ensemble experiments are previously possible using ApoE nanolipoprotein particles (NLPs) that require careful biochemical purifications[27,29,30]. Using engineered ApoE lipoprotein, NLPs could be produced with diameters from 10–30 nm[40,41]. Thus, these NLPs have been successfully used to dissect the molecular mechanism of SNARE-mediated fusion pores. However, the reconstituted materials are very polydisperse, with the majority of these particles showing diameters of 10–20 nm[29,30]. So, the reconstitution of large NLPs is technically challenging and limited by low yields. Using the approach reported in this work, we could readily trap fusion pores in a variety of conformations with much-improved stability than liposomes and non-circularized nanodiscs, which might be crucial for future structural interrogations. On this front, we also expect that our tools would promote mechanistic dissection of viral fusion that usually also could not be reconstituted in small nanodiscs[11].

Moreover, large cNDs allowed us to characterize complex protein-lipid interactions (Fig. 6). In conjunction with proximity labeling[42], these large nanodiscs will further extend our ability to detect weak and transient membrane protein interactomes, thereby conferring access to the membrane biochemical space unattainable in previous studies and opening up windows of pharmaceutical opportunities.

## Methods

**Chemicals and reagents**. 1,2-dioleoyl-sn-glycero-3-phosphocholine (PC), 1,2-dioleoyl-sn-glycero-3-phospho-l-serine (PS), 1-palmitoyl-2-oleoyl-sn-glycero-3-phosphoethanolamine (PE), 1,2-dioleoyl-sn-glycero-3-phospho-(1′-rac-glycerol) (PG), 1,2-dipalmitoyl-sn-glycero-3-phospho-ethanolamine-N-(7-nitro-2-1,3-benzoxadiazol-4-yl) (NBD-PE) and N-(lissamine rhodamine B sulfonyl)-1,2-dipalmitoyl-sn-glycero-3-phosphoethanolamine (rhodamine-PE) were obtained from Avanti Polar Lipids. Nitrilotriacetic acid (Ni$^{2+}$-NTA)-chelating Sepharose and Superose 6 increase 10/300 GL were purchased from GE Healthcare. 1,1′-Dioctadecyl-3,3,3′,3′-Tetramethylindocarbocyanine Perchlorate) (Dil), N,N′-Dimethyl-N-(Iodoacetyl)-N′-(7-Nitrobenz-2-Oxa-1,3-Diazol-4-yl)Ethylenediamine (IANBD amide) and Oregon Green$^{TM}$ 488 (OG) maleimide were obtained from Thermo-Fisher. All other chemicals were acquired from Sigma.

**Plasmids**. pET28a-MSP1D1 was a gift from Dr. Steven Sligar[7]. pET28a-NW30 and pET28a-NW50 were gifts from Dr. Gerhard Wagner[11]. All other constructs in this work were made using the In-Fusion® HD Cloning Kit (Takara Bio USA). Spy-Catcher002 was synthesized as a gBlocks gene fragment (IDT), and inserted together with SpyTag002 into pET28a-MSP1D1, yielding pET28a-spMSP1D1. pET28a-spNW30 and pET28a-spNW50 were made by replacing MSP1D1 in pET28a-spMSP1D1 with NW30 and NW50 amplified from pET28a-NW30 and pET28a-NW50. pET28a-spNW80 and pET28a-spNW100 were generated by inserting NW30 and NW50 into pET28a-spNW50. pET28a-spNW15 and pET28a-spNW25 were built by truncation from pET28a-spNW30. spMSP protein sequences are described in Supplementary Table I. All the spMSP plasmids have been deposited in Addgene: spMSP1D1 (ID: 173482), spNW15 (ID: 173483), spNW25 (ID: 173484), spNW30 (ID: 173485), spNW50 (ID: 173486), spNW80 (ID: 173487), spNW100 (ID: 173488).

Full-length cpx2 and the fragment encoding the cytoplasmic domain of syt1 were cloned into pGEX-4T[43]. Full-length EIIA$^{Glc}$ was cloned into pBad33 with a C-terminal His-tag[33]. For site-directed labeling, native cysteines were substituted with alanine. Then, single cysteines were introduced at Phe234 of syt1, Lys134 of cpx2, Thr96 of EIIA$^{Glc}$.

**Proteins**. syt1, cpx2, EIIA$^{Glc}$, MSP1D1, and SNAREs were expressed in BL21 STAR$^{TM}$ (DE3) and purified using GSTrap and Ni$^{2+}$-NTA columns[20,25,44–47]. To produce spMSP1D1,spNW15, spNW25, spNW30, spNW50, spNW80, and spNW100, plasmids were transformed into BL21 STAR$^{TM}$ (DE3) cells that were grown in LB supplemented with Km (50 mg/ml) to OD$_{600}$ ~ 0.7. Protein expression was induced with 0.2 mM IPTG at 16 °C overnight. Bacteria were harvested by centrifugation at 3450 × g for 20 min, resuspended in buffer A (50 mM Tris-HCl (pH 8),100 mM NaCl, 5% glycerol, 2 mM β-mercaptoethanol), and lysed on ice using a Branson cell disrupter (60% duty cycle, 45 secs). Cell lysates were clarified by centrifugation at 12,000 × g for 45 mins. The supernatants were loaded onto a 1 ml Ni$^{2+}$-NTA column (GE Healthcare), followed by extensive wash (20 column volume) using buffer B (50 mM Tris-HCl (pH 8), 20 mM Imidazole, 400 mM NaCl, 5% glycerol, 2 mM β-mercaptoethanol). Proteins were eluted in buffer C (50 mM Tris-HCl (pH 8), 500 mM Imidazole, 400 mM NaCl, 5% glycerol, 2 mM β-mercaptoethanol), desalted in buffer A using PD MiDiTrap G-25 (GE Healthcare), and stored at −80 °C.

**Fluorescent labeling of proteins**. Purified syt1, cpx2, and EIIA$^{Glc}$ were desalted using Zeba Spin columns (Thermo Fisher) in buffer D (50 mM Tris-HCl, pH 8, 100 mM NaCl, 5% glycerol) and labeled with a 3-fold excess of IANBD amide or OG maleimide in the presence of TCEP (0.2 mM) at room temp for 2 h. Free dyes were removed by passing through Zeba Spin columns in buffer A.

For fluorescence fluctuation spectroscopy experiments, spMSP1D1, spNW30, and spNW50 were desalted using Zeba Spin columns in PBS buffer and labeled with a 3-fold excess of fluorescein isothiocyanate (FITC) at room temp for 2 h. Free dyes were removed by passing through Zeba spin columns in buffer A.

**Nanodiscs**. A step-by-step protocol describing nanodisc reconstitution can be found at Protocol Exchange[48]. To optimize the condition for cND reconstitution, spMSPs were incubated with PC lipids at the indicated ratio in buffer A containing 0.05% DDM. To incorporate v-SNARE into nanodiscs, v-SNARE, MSPs, and PC lipids, were mixed at a ratio of 2:5:60 for spMSP1D1, 2:5:600 for spNW25, and 2:5:2000 for spNW50 in buffer A containing 0.05% DDM. For lipid mixing assays, v-SNARE nanodiscs were reconstituted with 12% PE, 40% PS, 45% PC, 1.5% NBD-PE, and 1.5% rhodamine-PE. To monitor the binding of syt1 to lipids via fluorescence spectrometry, nanodiscs were made using 70% PC and 30% PS. To study the interaction of cpx2 and EIIA$^{Glc}$ via FRET, nanodiscs were made using 1% Dil, 69% PC, and 30% PS or PG. Samples were kept on ice for 30 min, and detergents were slowly removed with BioBeads (1/3 volume) and gentle shaking (4 °C, overnight). cNDs were purified by SEC in Superose 6 10/300 in buffer A and stored at −80 °C.

**Crosslinking experiments**. cNDs (0.5 μM) were desalted in PBS using Zeba Spin columns and incubated with the indicated concentration of ethylene glycol bis(succinimidyl succinate) (EGS, ThermoFisher) at room temperature for 1 h. Reactions were quenched by the addition of 100 mM Tris-HCl (pH 8), and samples were subject to SDS-PAGE analysis. Image Lab software (BioRad) was used to collect data and Image Studio$^{TM}$ Lite Ver 5.2 (LI-COR) to analyze data.

**Negative stain electron microscopy**. NDs (10 μg/ml) were applied onto Formvar/carbon-coated copper grids (01754-F, Ted Pella, Inc.) that were glow discharged (15 mA, 25 s) using PELCO easiGlow$^{TM}$ (Ted Pella, Inc). After 30 s, samples were stained with 0.75% uranyl formate for 1 min. Grids were imaged on a Thermo-Fisher Science Tecnai G2 TEM (100 kV) equipped with a Veleta CCD camera (Olympus). TIA (FEI) software was used to collect data and ImageJ2 Fiji to analyze data.

**Mass spectrometry**. Protein samples were incubated with trypsin at 37 °C overnight. The resulting peptide samples were injected for inline pepsin digestion and the resulting peptides were identified using tandem MS (MS/MS) with an Orbitrap mass spectrometer (Q Exactive, ThermoFisher). Following digestion, peptides were desalted on a C8 trap column and separated on a 1 h linear gradient of 5–40% B (A is 0.3% formic acid and B is 0.3% formic acid 95% CH$_3$CN). Product ion spectra were acquired in a data-dependent mode such that the 10 most abundant ions were selected for the product ion analysis by higher-energy collisional dissociation between survey scans. Following MS2 acquisition, the precursor ion was excluded for 16 seconds. The resulting MS/MS data files were submitted to Proteome Discoverer (Thermo, version 2.5.0.400) for database searches and peptide identification. The precursor and fragment mass tolerances were set to 10 ppm and the false discovery rate was set to 5%. The MS/MS spectra identifying peptides were verified by manual inspection. The Proteome Discoverer results were used as input for skyline (version 20.2.0.343) to quantify peak areas.

**Fusion assays**. For the cargo release assay, t-SNARE vesicles encapsulated with glutamate or fluorescence dextran were prepared as described previously[20,25]. The reaction was initiated by incubation of v-SNARE nanodiscs (0.5 μM) and t-SNARE vesicles (1 μM) in reconstitution buffer (20 mM Tris-HCl, pH 7.5, 100 mM NaCl) at 37 °C for 30 mins. For the glutamate release assay, the glutamate sensor iGluSnfR[49,50] (0.1 μM) was added to the reaction mixture. For the dextran release assay, samples were centrifuged at 50,000 × g for 30 min, and the supernatants were carefully collected for

further analysis. The fluorescence of iGluSnFR and dextran was quantified using a Synergy H1M plate reader with excitation at 460 nm and emission at 530 nm. The percentages of cargo release were determined by normalization of data to the maximal release after the addition of 0.5% DDM to each sample.

For the lipid mixing assay, v-SNARE vesicles were prepared by detergent removal[20,25]. Briefly, purified synaptobrevin-2 (syb-2) were incubated with lipids (12% PE, 40% PS, 45% PC, 1.5% NBD-PE and 1.5% rhodamine-PE) in the reconstitution buffer supplemented with 0.1% DDM. Detergents were slowly removed by addition of Biobeads (1/3 volume) and gentle shaking (4 °C, overnight). Liposomes harboring v-SNAREs were extruded through an Avanti extruder with 200 nm filter and further purified using PD MiDiTrap G-25 (GE Healthcare) in reconstitution buffer. The lipid mixing assays were performed by incubation of v-SNARE vesicles or nanodiscs bearing the FRET reporter (NBE-PE and Rhodamine-PE) with t-SNARE vesicles in the presence of C2AB (1 μM) and $Ca^{2+}$ (0.5 mM). The NBD signal was monitored for an additional 1 h using a Synergy H1M plate reader with excitation at 460 nm and emission at 530 nm. The efficiencies of membrane fusion were determined by normalization of data to the maximal release after the addition of 0.5% DDM to each sample. BioTek Gen5 software was used to collect data and Graphad 8 to analyze data.

**Fluorescence spectroscopy**. NBD- and OG-labeled proteins (10 nM) were incubated with nanodiscs at the indicated concentrations in reconstitution buffer. The fluorescence spectrum of samples was collected on a Synergy H1M plate reader with excitation at 460 nm and emission from 500–650 nm.

**Fluorescence fluctuation spectroscopy**. spMSPs were labeled with FITC and then used for cND reconstitutions. The molecular brightness of FITC-labeled spMSPs and cNDs (100 pM) in reconstitution buffer were quantified as counts per second per molecule (cpsm) via photon counting histogram using an ISS Alba v5 laser scanning microscope (ISS, Inc.). Samples were excited using a pulsed 488 nm laser, and photon counts were detected with an avalanche photodiode (APD) (Excelitas, SPCM-AQRH-15). Data were analyzed using the ISS vista vision software.

**Other methods**. SDS-PAGE and Native PAGE electrophoresis were performed using 4–15% TGX stain-free™ protein gels (Bio-Rad) and imaged using the Image Lab software on a GelDoc Go system (Biorad). Size-exclusion chromatography (SEC) was carried out on an AKTA pure 25 L using Superose 6 10/300 (GE Healthcare) in 50 mM Tris-HCl, pH 8, 100 mM NaCl, 5% Glycerol[45,46]. Unicorn 7 software was used to collect data and Graphad 8 to analyze data. Density gradient flotation was performed on a Beckman Optima XE-90 Ultra using an Accudenz step gradient (Accurate Chemical & Scientific Corporation)[20,25].

**Reporting summary**. Further information on research design is available in the Nature Research Reporting Summary linked to this article.

## Data availability
Gel images and fluorescence spectroscopy data are provided in the Source Data file and all other data are included in the paper. Source data are provided with this paper.

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

## Acknowledgements

We thank Drs. Naomi Kamasawa and Debby Guerrero-Given from the imaging center at the Max Planck Florida Institute for Neuroscience for assist in EM. We would also like to thank Dr. Franck Duong for the plasmid of pBad33-EIIA^Glc and Dr. Edwin Chapman for the plasmid pGEX4T-syt1 and pGEX4T-cpx2. This work was made possible by support from the Scripps Research Institute and the NIH Director's New Innovator Award (DP2GM140920 to B.H.).

## Author contributions

Z.S. and R.Q. performed experiments and data analysis for biochemical assays and negative stain EM. S.N., T.S., and G.P. carried out mass spectrometry and analyzed the data. B.H. conceived the project and wrote the paper with input from all authors.

## Competing interests

The authors declare no competing interests.
