## [Peer Review File · Nature Communications]

Reviewers' Comments:

Reviewer #1:

Remarks to the Author:

This manuscript uses SpyCatcher and SpyTag fusions to membrane scaffold proteins (MSP) to design circularized nanodiscs with improved expression and bigger sizes. It solves a number of challenges with other circularized nanodisc systems. The overall approach builds on prior work and will be of great interest to the large community of nanodisc users.

My primary concern is that the authors have not mentioned the public availability of the plasmids. For this to be a useful resource of the community, the authors need to deposit the plasmids in Addgene or another public repository, which has been done for most other MSP variants.

Work on in vivo circularized MSPs from Hagn and coworkers [ref. 22] should be mentioned in the introduction as it is very similar.

Did the authors ever try smaller MSP variants?

The number of lipids found for the spMSP1D1 and the mass is far higher than conventional MSP1D1. Can the author explain the anomalously high masses?

Also, some of the nanodiscs seem to show quite polydisperse SEC traces compared to the Sligar and Wagner papers. Can the authors comment more on how these cNDs compare with other cND designs and conventional MSPs in terms of the nanodisc assembly?

For the binding studies in Figure S5, the binding is plotted as a function of nanodisc concentration. Would the results change if they were plotted as a function of lipid concentration, which is much higher in the 50 nm nanodiscs? I'm wondering if the difference in affinity is merely a function of higher lipid concentrations.

Is the "protein superglue" term standard in the literature? It seems unnecessary and confusing. I would recommend changing it to just SpyCatcher/SpyTag or something like that.

Reviewer #2:

Remarks to the Author:

The authors describe an improved method to make circularized nanodiscs that would potentially be very useful for many researchers. The idea for linking the ends is very simple: fuse SpyCatcher and SpyTag to the N- and C-termini of an ApoA1 based membrane scaffold protein (MSP), respectively, and the circularized scaffold is produced by bacteria. Compared to previous cND protocols, the new approach appears to be much simpler and easier.

The authors demonstrate the use of the new cNDs in studying SNARE-mediated membrane fusion and protein-lipid interactions.

Major issues:

1) The impact of this type of work depends very much on the context. Although the method focuses on circularized NDs, there are other ways to make ~30 nm or larger nanodiscs which are not mentioned. In particular, nanolipoprotein particles (NLPs) were developed by Chromy et al. (*J Am Chem Soc* 129:14348–14354, 2007), characterized in detail in Blanchette et al. (*J Lipid Res* 49, 1420–1430, 2008), and used in studies of SNARE-mediated fusion pores in bulk by Bello et al. (*Langmuir* 32, 3015–3023, 2016, doi:10.1021/acs.langmuir.6b00245) and in single pore studies by Wu et al. (*Elife* 6, e22964, 2017, doi:10.7554/eLife.22964). These need to be cited and the ease of preparation, size distribution, etc of the new cNDs compared to NLPs.

Wu et al used NLPs to test the role SNARE copy numbers in expanding fusion pores in a single-pore study (Wu et al. *eLife* 2017 mentioned above), before the self-cited study by the authors (ref.

12) reported similar findings. What is the advantage of circularized NDs over NLPs?? Is there any?

More broadly, previous work is conspicuously ignored throughout the text, with some claims that are simply not correct. For example, 1st paragraph in p 9 claims "Here, we first showcased the power of large nanodiscs in the reconstitution of fusion pore dilation (Fig. 5), which previously is challenging and could only be achieved at the single-molecule level due to the low yields of the sortase-mediated protein ligation method." Bello et al. mentioned above used NLPs to interrogate fusion pore sizes in a bulk dextran release assay, so they must have been able to prepare a sufficient quantity of SNARE-reconstituted NLPs.

Even the use of liposomes as tools in studies of membrane protein function and membrane fusion studies is not mentioned. What is the advantage of nanodiscs, in particular large ones, compared to liposomes?

To be fair, preparation of NLPs is not trivial and aggregation is a major issue. So the new method could in fact present a major improvement, but this needs to be discussed and justified, and NLPs and alternative ways of making large NDs need to be mentioned.

2) Bottom of p5. The authors were "amazed" that spNW30 produced 32 nm NDs, as opposed to the 15 nm as expected from the sortase cNDs. The authors should quantify the number of MSP copies per disc. An ideal way to do this would be to make cNDs using spNW30 labeled with an organic dye, and additionally including fluorescently labeled lipids in the NDs. Attaching these NDs sparsely onto a coverslip, the distribution of the lipid to protein ratio could be quantified using fluorescence microscopy (ideally TIRFM). A similar single liposome-ND fusion approach was used by Bao et al (ref 12), so the authors should be knowledgeable in such approaches. With proper calibration, they can also obtain absolute numbers for the copies of MSP and lipids per disc.

This would also be a very useful approach to quantify the size distribution of the 80 and 100 nm discs (using fluo intensity of lipids in NDs on a coverslip).

3) it is claimed that circularization occurred spontaneously in bacteria, producing almost exclusively monomers. It is surprising that dimers, trimers, etc are not observed. Why is that? The occurrence of higher order oligomers is expected to depend on the concentration of MSP. Can the authors try to express MSP at different levels in bacteria? (at least to test production of good quality MSP will be fine over some range of expression levels).

4) Since this manuscript is about a new approach to produce cNDs, the description of protein purification and cND production should be expanded. In particular, a schematic of the workflow and indication of critical steps would be very useful. The stability and storage conditions should also be explored and reported.

minor:

5) Figure 6 and S5B: Plot legend is missing in all of C-F. Plot color (black, blue, red) should be clarified in figure legend.

Figure 5A is not very clear. Add a panel showing liposomes and NDs are mixed together. Explain how release is measured.

6) Statistical tests are missing to determine if differences between means are significant. Put fig S4 (negative controls) together with fig 5.

7) Fig S5 (and in general). How are ND concentrations reported? Number of NDs/volume? Or is it the total lipid that is reported (total number lipids/volume). The distinction is very important for interpretation of binding data.

8) There are many abbreviations that are not defined. Many captions are not detailed enough.

Reviewer #3:

Remarks to the Author:

In their paper, "One-step construction of circularized nanodiscs using protein superglue," Zhang and co-authors describe a method for fabricating lipid-containing nanodiscs based on established membrane scaffold proteins using the established SpyCatcher/SpyTag system. The nanodiscs can be fabricated in various sizes beyond those produced by previously established methods. In addition, the nanodiscs can be purified in a simple, one-step process. In particular, the significantly increased size of the nanodiscs can be useful in contexts where membranes are studied and as illustrated by the authors, for example, for the binding of peripheral membrane proteins. The approach presented in this work therefore has the potential to advance the field.

Some important details, such as the detailed protein sequences, are missing and would prevent others from reproducing the work independently. These would need to be included to consider publishing the paper.

The authors cite and explain previously published systems that utilize the enzyme sortase for the cyclization of membrane scaffold proteins. In their work, the authors use the SpyCatcher/SpyTag system to enable this cyclization, however, they do not cite any of the previous work that used this type of cyclization, which has been used for other proteins.

For example

Schoene, C., Fierer, J.O., Bennett, S.P. and Howarth, M. (2014), SpyTag/SpyCatcher Cyclization Confers Resilience to Boiling on a Mesophilic Enzyme. *Angew. Chem. Int. Ed.*, 53: 6101-6104. <https://doi.org/10.1002/anie.201402519>

Wang, J., Wang, Y., Wang, X., Zhang, D., Wu, S., & Zhang, G. (2016). Enhanced thermal stability of lichenase from *Bacillus subtilis* 168 by SpyTag/SpyCatcher-mediated spontaneous cyclization. *Biotechnology for biofuels*, 9, 79. <https://doi.org/10.1186/s13068-016-0490-5>

Si M, Xu Q, Jiang L, Huang H (2016) SpyTag/SpyCatcher Cyclization Enhances the Thermostability of Firefly Luciferase. *PLoS ONE* 11(9): e0162318. <https://doi.org/10.1371/journal.pone.0162318>

Schoene, C., Bennett, S. P., & Howarth, M. (2016). SpyRing interrogation: analyzing how enzyme resilience can be achieved with phytase and distinct cyclization chemistries. *Scientific reports*, 6, 21151. <https://doi.org/10.1038/srep21151>

Sun XB, Cao JW, Wang JK, Lin HZ, Gao DY, Qian GY, Park YD, Chen ZF, Wang Q.

SpyTag/SpyCatcher molecular cyclization confers protein stability and resilience to aggregation. *N Biotechnol.* 2019 Mar 25;49:28-36. doi: 10.1016/j.nbt.2018.12.003.

These papers contain useful information for a discussion on the effects observed. Especially the paper from Sun et al also reports resilience to aggregation, which is similar to the decreased tendency of protein constructs of this work to be not soluble.

Some details are rather difficult to understand or are missing information to understand. For example,

- Two non-circularized variants are mentioned (EQ and DA). I assume those are E -> Q and D -> A mutants? At which position in the protein? There are no citations to earlier works, which explain these mutants.
- Citations for the specific MSP used as well as for the NW30 and NW50 protein are missing in the main text.
- Figure 2 needs more explanation. Information on the standard set of proteins mentioned can not be found in the figure legend nor the methods part. This information is relevant to understand the apparent difference in the size of the proteins between Figure 1B (SDS-PAGE) and Figure 2 top row (proteins with no lipid). The authors should comment on these apparent differences.
- Rhodamine-PE, the abbreviation PE is not explained.
- Details on the cloning are missing. It is well accepted upon fusion of proteins with tags such as the SpyCatcher/SpyTag, linker sequences can be important for protein function. Providing the full protein sequence for all fusion proteins would solve this problem.

- For figures S1, S2 and S4 error bars are depicted, but no info on the number of samples is given.

A technical question: in Figure S2 the protein content of the cNDs is calculated via their absorption at 280 nm. The lipids per cND are determined via the Rhodamine fluorescence. Based on 0.5% Rhodamine-PE doping roughly 1, 3 and 9 molecules per cND would result from the numbers presented, while there are two protein molecules in the cND. Since Rhodamine also has an absorption at 280 (please provide a spectrum/molecular extinction coefficient), how does that relate to the absorption of the proteins? Has this absorption been subtracted?

Reviewer #1 (Remarks to the Author):

This manuscript uses SpyCatcher and SpyTag fusions to membrane scaffold proteins (MSP) to design circularized nanodiscs with improved expression and bigger sizes. It solves a number of challenges with other circularized nanodisc systems. The overall approach builds on prior work and will be of great interest to the large community of nanodisc users.

Response: We are very grateful for the enthusiasm of this reviewer in our work. We hope that the reviewer finds the revised manuscript satisfactory.

1. My primary concern is that the authors have not mentioned the public availability of the plasmids. For this to be a useful resource of the community, the authors need to deposit the plasmids in Addgene or another public repository, which has been done for most other MSP variants.

Response: We totally agree with the reviewer, and we have now deposited our plasmids in Addgene. In the revised manuscript, we have made the following changes:

“spMSP protein sequences are described in Supplementary Table I. All the spMSP plasmids have been deposited in Addgene: spMSP1D1 (ID: 173482), spNW15 (ID: 173483), spNW25 (ID: 173484), spNW30 (ID: 173485), spNW50 (ID: 173486), spNW80 (ID: 173487), spNW100 (ID: 173488).”

2. Work on in vivo circularized MSPs from Hagn and coworkers [ref. 22] should be mentioned in the introduction as it is very similar.

Response: We agree with the reviewer. In the revised manuscript, we have now discussed the work by Hagn and coworkers in the introduction as the following:

“To address these problems, previous studies have also attempted using split inteins to circularize MSPs in cells¹³. However, purification of the circularized MSPs requires multiple chromatography steps and thus time-consuming. Moreover, it is unclear if the split-intein-based approach is amenable to building large cNDs with diameters over 30 nm.”

3. Did the authors ever try smaller MSP variants?

Response: The smaller MSP variants developed by Hagn et al. (2013) were already quite homogeneous. Thus, we think they do not need to be further improved. The circularization approach described in our manuscript was aimed to improve the construction of large nanodiscs with diameters over 20 nm. In the revised manuscript, we have explained this point as the following:

“systematic truncation of MSP1D1 could yield stable and monodisperse small nanodiscs with diameters from 6 to 9 nm¹⁰.”

4. The number of lipids found for the spMSP1D1 and the mass is far higher than conventional MSP1D1. Can the author explain the anomalously high masses?

Response: We think the difference could be caused by the increased size of spMSP1D1 vs. MSP1D1 nanodiscs (11 vs. 9 nm). In the original paper from the Sligar group, the size of MSP1D1 nanodiscs characterized by gel-filtration was close to β -Amylase (200 kDa) and contains 62 lipids per MSP (equivalent of 124 per nanodisc) (Bayburt et al., Nano Lett., 2002; Denisov et al., 2004 JACS). In contrast, our spMSP1D1 nanodiscs were 280 kDa, containing ~250 lipids per nanodisc calculated based on gel-filtration (Fig. 2) and ~188 lipids by quantification using fluorescent lipids (Fig. S3). Given the MW of spMSP1D1 is 39 kDa vs. 23 kDa of MSP1D1, the difference in mass by lipids is 48 kDa, and the difference in the number of lipids is about 64-126 copies. This increase in the mass and number of lipids is consistent with the 2 nm expansion in the diameter of spMSP1D1 vs. MSP1D1 nanodiscs. We suspect that the

addition of SpyCatcher-SpyTag in spMSP1D1 nanodiscs is the underlying cause of this difference, and we will further characterize this observation in future studies.

In the revised manuscript, we have discussed this issue and made the following changes: “Our work is consistent with previous studies showing that MSPs play a key role in determining the characteristics of nanodiscs^{6, 7, 13}. We observed that the addition of SpyCatcher-SpyTag in nanodisc changed its physical and chemical properties. The size and mass of spMSP1D1 nanodiscs (11 nm and 280 kDa) were higher than that of the MSP1D1 nanodiscs (9 nm and 200 kDa). In addition, the stability of circularized nanodiscs was markedly enhanced as compared to non-circularized nanodiscs and liposomes (Fig. S6). Therefore, the properties of nanodiscs could be fine-tuned by further engineering of MSPs.”

5. Also, some of the nanodiscs seem to show quite polydisperse SEC traces compared to the Sligar and Wagner papers. Can the authors comment more on how these cNDs compare with other cND designs and conventional MSPs in terms of the nanodisc assembly?

Response: For the small nanodisc (MSP1D1) in the Sligar and Wagner papers (Denisov et al., 2004 JACS; Hagn et al., 2013 JACS), the homogenous SEC profiles with sharp peaks were the second fractionation of nanodisc samples isolated from a first fractionation. In our original manuscript, the traces shown in Fig. 2 were the SEC profiles of the first fractionation of reconstituted nanodiscs, thus having peaks corresponding to aggregates, free MSPs, and particles of different sizes. We decided to show these data, so readers will know what they should expect from the first fractionation of reconstituted nanodiscs. In addition, it is known that nanodisc reconstitution requires optimization of the MSP: lipid ratio. Decreasing or increasing this ratio will result in polydisperse particles (Fig. 2), consistent with previous studies from the Sligar group (Bayburt et al., Nano Lett., 2002; Denisov et al., 2004 JACS).

For the large nanodiscs, we could not find SEC profiles from the Wagner paper (Nasr et al., 2017 Nat Methods), so we cannot compare. Nevertheless, we agree with the reviewer that we should compare our work with previous studies. To do so, we have now re-injected the purified nanodiscs for a second SEC analysis (new Fig. S2). The results support that our cNDs are quite monodisperse, in agreement with the negative-stain EM studies (Fig. 3)

In the revised manuscript, we have included these data in the supplementary figures (new Fig. S2) and made the following changes: “The reconstituted materials were separated by SEC using a Superose 6 column (Fig. 2), and fractions from the main peaks were analyzed by negative stain EM (Fig. 3) and a second SEC (Fig. S2). Upon incorporation of optimal amounts of lipids, monodisperse protein-lipid particles formed with the three circularized MSPs (Fig. 2, 3 and Fig. S2). Increasing or decreasing the lipid-protein ratios resulted in reduced yields of the main peaks and the formation of multiple species (Fig. 2), similar to previous studies of nanodisc assembly using non-circularized MSPs^{6, 7}.”

6. For the binding studies in Figure S5, the binding is plotted as a function of nanodisc concentration. Would the results change if they were plotted as a function of lipid concentration, which is much higher in the 50 nm nanodiscs? I'm wondering if the difference in affinity is merely a function of higher lipid concentrations.

Response: We agree with the reviewer. To test if our results were due to higher lipid concentrations when using large nanodiscs, we have performed more titrations and plotted our data as a function of lipid concentration. The results are similar to plots as a function of nanodisc concentration (new Fig. S7A and B), suggesting that cooperative binding of syt1 to lipids only occurs on large nanodiscs.

In the revised manuscript, we have made the following changes: “These observations were not caused by the higher amount of lipids using large spNW50 nanodiscs, as plotting these data as a function of lipid concentrations yields similar results (Fig. S7B).”

7. Is the “protein superglue” term standard in the literature? It seems unnecessary and confusing. I would recommend changing it to just SpyCatcher/SpyTag or something like that.

Response: We found this nomenclature from several previous papers by the Howarth group (Veggiani et al., 2014 Trends Biotechnol; Wieduwild and Howarth, 2018 Biomaterials), so we used it in our original manuscript. We agree that it could be confusing for people that are not familiar with the SpyCatcher-SpyTag technology. In the revised manuscript, we have replaced the term ‘protein superglue’ with ‘SpyCatcher-SpyTag’.

Reviewer #2 (Remarks to the Author):

The authors describe an improved method to make circularized nanodiscs that would potentially be very useful for many researchers. The idea for linking the ends is very simple: fuse SpyCatcher and SpyTag to the N- and C-termini of an ApoA1 based membrane scaffold protein (MSP), respectively, and the circularized scaffold is produced by bacteria. Compared to previous cND protocols, the new approach appears to be much simpler and easier. The authors demonstrate the use of the new cNDs in studying SNARE-mediated membrane fusion and protein-lipid interactions.

Response: We very much appreciate the positive feedback from this reviewer. We hope that the reviewer finds the revised manuscript satisfactory.

Major issues:

1) *The impact of this type of work depends very much on the context. Although the method focuses on circularized NDs, there are other ways to make ~30 nm or larger nanodiscs which are not mentioned. In particular, nanolipoprotein particles (NLPs) were developed by Chromy et al. (J Am Chem Soc 129:14348–14354, 2007), characterized in detail in Blanchette et al. (J Lipid Res 49, 1420–1430, 2008), and used in studies of SNARE-mediated fusion pores in bulk by Bello et al. (Langmuir 32, 3015-3023, 2016, doi:10.1021/acs.langmuir.6b00245) and in single pore studies by Wu et al. (Elife 6, e22964, 2017, doi:10.7554/eLife.22964). These need to be cited and the ease of preparation, size distribution, etc of the new cNDs compared to NLPs.*

Response: We agree with the reviewers that we should extend our discussion to NLPs, and have made the following changes in the revised manuscript:

“Ensemble experiments are previously possible using ApoE nanolipoprotein particles (NLPs) that require careful biochemical purifications²⁸⁻³⁰. Using engineered ApoE lipoprotein, NLPs could be produced with diameters from 10-30 nm^{38, 39}. Thus, these NLPs have been successfully used to dissect the molecular mechanism of SNARE-mediated fusion pores. However, the reconstituted materials are very polydisperse, with the majority of these particles showing diameters of 10-20 nm^{29, 30}. So, purification of large NLPs encased using ApoE is technically challenging and limited by low-yields.”

1a. *Wu et al used NLPs to test the role SNARE copy numbers in expanding fusion pores in a single-pore study (Wu et al. eLife 2017 mentioned above), before the self-cited study by the authors (ref. 12) reported similar findings. What is the advantage of circularized NDs over NLPs?? Is there any?*

Response: NLPs are quite polydisperse and tend to aggregate. As shown in the papers mentioned above by the reviewer, NLPs usually contain a mixture of particles with diameters ranging from 10-30 nm. In the studies by Bello et al. and Wu et al., careful optimization and purification are thus needed to produce monodisperse NLPs. Moreover, reconstitution of large

NLPs using ApoE is difficult because the majority of particles are within 10-20 nm and require a significant amount of work to isolate them from small particles. In contrast, NDs have much-improved homogeneity and stability, making them the most widely used system for structural and functional studies of membrane proteins in a lipid environment. However, production of large NDs is limited by low yields and time-consuming procedures via the sortase-mediated protein ligation method. Using our approach, we could easily expand the size of nanodiscs from 10 to 50 nm with much better homogeneity than NLPs (Fig. 3 and new Fig. S2). Therefore, we suspect that our large circularized NDs could markedly alleviate the technical barrier for reconstitution of complex membrane systems that require expanded lipid environment. Nevertheless, we agree with the reviewers that we should discuss our work in the context of other lipid nanoparticles and have made the changes in the revised manuscript as described above in response to point #1.

1b. More broadly, previous work is conspicuously ignored throughout the text, with some claims that are simply not correct. For example, 1st paragraph in p 9 claims "Here, we first showcased the power of large nanodiscs in the reconstitution of fusion pore dilation (Fig. 5), which previously is challenging and could only be achieved at the single-molecule level due to the low yields of the sortase-mediated protein ligation method." Bello et al. mentioned above used NLPs to interrogate fusion pore sizes in a bulk dextran release assay, so they must have been able to prepare a sufficient quantity of SNARE-reconstituted NLPs.

Response: We apologized that we did not discuss the work by Bello et al. in our initial manuscript, as we mainly focused on circularized nanodiscs. In the revised manuscript, we have corrected our statements, extended our discussion into NLPs and have made the following changes:

"Here, we first showcased the power of circularized large nanodiscs in the reconstitution of fusion pore dilation (Fig. 5). We previously could only do so at the single-molecule level due to the low yields of the sortase-mediated protein ligation method. Ensemble experiments are previously possible using ApoE nanolipoprotein particles (NLPs) that require careful biochemical purifications²⁷⁻²⁹. Using engineered ApoE lipoprotein, NLPs could be produced with diameters from 10-30 nm^{38, 39}. Thus, these NLPs have been successfully used to dissect the molecular mechanism of SNARE-mediated fusion pores. However, the reconstituted materials are very polydisperse, with the majority of these particles showing diameters of 10-20 nm^{29, 30}. So, purification of large NLPs encased using ApoE is technically challenging and limited by low-yields."

1c. Even the use of liposomes as tools in studies of membrane protein function and membrane fusion studies is not mentioned. What is the advantage of nanodiscs, in particular large ones, compared to liposomes?

Response: We agree that the use of liposomes as tools in studies of membrane proteins and membrane fusion should be discussed. Liposomes are sealed vesicles, whereas nanodiscs are accessible from either side of the membrane. Conferred by this advantage in accessibility, we could use planar lipid bilayer electrophysiology to characterize single fusion pores. In addition, it is difficult to study nascent fusion pores using liposomes, as the pore would rapidly dilate when the two opposing lipid bilayers are fused. However, fusion pores could be trapped at distinct states using nanodiscs with different diameters, as the rigid framework of nanodiscs restricts the expansion of the pore.

Moreover, as compared to nanodiscs, liposomes were quite polydisperse, tend to precipitate, and thus unstable. Therefore, structural and biophysical studies of membrane proteins and membrane fusion in liposomes are much more difficult than nanodiscs. In the revised manuscript, we have discussed the use of liposomes in studies of membrane fusion and

compared the stability of nanodiscs vs. liposomes in the new Fig. S6. Together, we have made the following changes:

“Liposome-based reconstitution approach could study pore dilation²⁵, but it is difficult to parse out nascent fusion pores from dilated ones²⁶⁻²⁹.”

“we found that these cNDs exhibited much-improved stability than liposomes or non-circularized nanodiscs for the reconstitution of membrane fusion (Fig. S6).”

“we could readily trap fusion pores in a variety of conformations with much-improved stability than liposomes and non-circularized nanodiscs, which might be crucial for future structural interrogations.”

1d. To be fair, preparation of NLPs is not trivial and aggregation is a major issue. So the new method could in fact present a major improvement, but this needs to be discussed and justified, and NLPs and alternative ways of making large NDs need to be mentioned.

Response: We thank the reviewer for deeming that our work could present a major improvement. Also, we agree with the reviewer that we have been too focused on circularized nanodiscs in our original manuscript. In the revised manuscript, we have discussed our work in the context of NLPs and liposomes as described above in response to point #1a-c.

2) Bottom of p5. The authors were "amazed" that spNW30 produced 32 nm NDs, as opposed to the 15 nm as expected from the sortase cNDs. The authors should quantify the number of MSP copies per disc. An ideal way to do this would be to make cNDs using spNW30 labeled with an organic dye, and additionally including fluorescently labeled lipids in the NDs. Attaching these NDs sparsely onto a coverslip, the distribution of the lipid to protein ratio could be quantified using fluorescence microscopy (ideally TIRFM). A similar single liposome-ND fusion approach was used by Bao et al (ref 12), so the authors should be knowledgeable in such approaches. With proper calibration, they can also obtain absolute numbers for the copies of MSP and lipids per disc. This would also be a very useful approach to quantify the size distribution of the 80 and 100 nm discs (using fluo intensity of lipids in NDs on a coverslip).

Response: The NW30 construct was initially designed to form 30 nm nanodiscs, as it contains three duplicates of MSP1D1 (Nasr et al., 2017 Nat Methods). However, the sortase-mediated ligation approach could only produce 15 nm nanodiscs using NW30. Thus, we were very excited that we could successfully produce the expected 30 nm nanodiscs using the SpyCatcher-SpyTag approach. We agree with the reviewer that we should characterize the copy number of MSP in our new nanodiscs.

As a new lab started during the COVID-19 pandemic, our TIRF system is unfortunately still under construction, and we do not have access to other TIRF microscopes. We thus first used crosslinking to characterize the copy number of MSP in nanodiscs, as described in the initial nanodisc paper by the Sligar group (Bayburt et al., 2002, Nano Lett). The results showed that cNDs are most likely formed by two copies of MSP (new Fig. S4A). In addition, we have quantified the number of lipids per cND using fluorescently labeled lipids (new Fig. S3). Finally, we used photon brightness counting via fluorescence fluctuation spectroscopy, further supporting that nanodiscs contained two copies of MSPs (new Fig. S4B).

In the revised manuscript, we have made the following changes:

“This difference was not due to the altered assembly of spNW30 nanodiscs, because biochemical crosslinking and fluorescence fluctuation spectroscopy experiments showed that spMSP1D1, spNW30, and spNW50 cNDs are all formed by two copies of MSPs (Fig. S4).”

“Crosslinking experiments

cNDs (0.5 μ M) were desalted in PBS using Zeba Spin columns and incubated with the indicated concentration of ethylene glycol bis(succinimidyl succinate) (EGS, ThermoFisher) at room temperature for 10 mins. Reactions were quenched by addition of 100 mM Tris-HCl (pH 8), and samples were subject to SDS-PAGE analysis.”

“Fluorescence fluctuation spectroscopy

spMSPs were labeled with FITC and then used for cND reconstitutions. The molecular brightness of FITC-labeled spMSPs and cNDs (100 pM) in reconstitution buffer were quantified as counts per second per molecule (cpsm) via photon counting histogram using an ISS Alba v5 laser scanning microscope (ISS, Inc.). Samples were excited using a pulsed 488 nm laser, and photon counts were detected with an avalanche photodiode (APD) (Excelitas, SPCM-AQRH-15). Data were analyzed using the ISS vistavision software”.

3) it is claimed that circularization occurred spontaneously in bacteria, producing almost exclusively monomers. It is surprising that dimers, trimers, etc are not observed. Why is that? The occurrence of higher order oligomers is expected to depend on the concentration of MSP. Can the authors try to express MSP at different levels in bacteria? (at least to test production of good quality MSP will be fine over some range of expression levels).

Response: We expressed spMSPs at 16 °C to make sure that most of the proteins are monomers. Our initial attempt expressing spMSPs at 37 °C resulted in more oligomers, as also pointed out by the reviewer. We think that this difference is two-fold by the lower temperature. First, protein diffusion is much slower at 16 °C, thus enforcing stronger inhibition on intermolecular than intramolecular ligation. Second, protein translation is less efficient, and as a consequence, decreasing the concentration of free spMSPs that could form the intermolecular covalent bond. In the revised manuscript, we have included the results of protein expressed at 37 °C and have made the following changes:

“We expressed the protein at 16 °C because more oligomers were formed at 37 °C (Fig. S1A).”

4) Since this manuscript is about a new approach to produce cNDs, the description of protein purification and cND production should be expanded. In particular, a schematic of the workflow and indication of critical steps would be very useful. The stability and storage conditions should also be explored and reported.

Response: In the revised manuscript, we have further expanded the description of protein purification and cND production. In addition, we have included a schematic diagram of the workflow to produce cNDs using our approach (new Fig. S8). We have characterized the stability and storage conditions of cNDs in the new Fig. S6. Together, we have made the following changes:

“In this manuscript, we presented a new method to ease the construction of cNDs (Fig. S8).”

“As a consequence, spMSPs exhibited greatly improved solubility and yields (Fig. 1 and Fig. S1), and only needed a single step for purification (Fig. S8).”

“the stability of cNDs was markedly enhanced as compared to non-circularized nanodiscs and liposomes (Fig. S6)”.

minor:

5) Figure 6 and S5B: Plot legend is missing in all of C-F. Plot color (black, blue, red) should be clarified in figure legend.

Figure 5A is not very clear. Add a panel showing liposomes and NDs are mixed together. Explain how release is measured.

Response: We apologized for these mistakes. In the revised manuscript, we have corrected them, added a panel of liposomes mixing with NDs in the new Fig. 5A, and made the following changes in the figure legends of Fig. 6 and S5B (the new Fig. S7C and D):

“(C-D) Fluorescence emission spectrum of NBD-labelled syt1 (10 nM) in the absence (black) or presence of 11 (C, blue) and 50 (D, red) nm cNDs (1 μM). (E-F) Fluorescence

emission spectrum of OG-labelled EIIA^{Glu} (10 nM) in the absence (black) or presence of Dil-labelled 11 (E, blue) and 50 (F, red) nm cNDs (10 μM). “

“(C-D) Fluorescence emission spectrum of OG-labelled cpx2 (10 nM) in the absence (black) or presence of Dil-labelled 11 (C, blue) and 50 (D, red) nm cNDs (10 μM)”.

6) *Statistical tests are missing to determine if differences between means are significant. Put fig S4 (negative controls) together with fig 5.*

Response: In the revised manuscript, we have performed statistical tests and combined Fig.S4 with Fig. 5 (new Fig. 5).

7) *Fig S5 (and in general). How are ND concentrations reported? Number of NDs/volume? Or is it the total lipid that is reported (total number lipids/volume). The distinction is very important for interpretation of binding data.*

Response: NDs concentrations were initially reported as the concentration of NDs/volume. So, the difference in the original Fig.S5A could be the higher lipid concentration in large NDs. To test this possibility, we have performed more titrations using small cNDs to achieve the same lipid concentration as the large cNDs, and plotted our data as a function of both ND and lipid concentrations in the new Fig. S7A-B. The cooperative binding of syt1 to lipids was still only observed using large cNDs. In the revised manuscript, we have included these data and made the following changes:

“These observations were not caused by the higher amount of lipids using large spNW50 nanodiscs, as plotting these data as a function of lipid concentrations yields similar results (Fig. S7B)”.

8) *There are many abbreviations that are not defined. Many captions are not detailed enough.*

Response: We apologized for these mistakes and have carefully edited our revised manuscript.

Reviewer #3 (Remarks to the Author):

In their paper, "One-step construction of circularized nanodiscs using protein superglue," Zhang and co-authors describe a method for fabricating lipid-containing nanodiscs based on established membrane scaffold proteins using the established SpyCatcher/SpyTag system. The nanodiscs can be fabricated in various sizes beyond those produced by previously established methods. In addition, the nanodiscs can be purified in a simple, one-step process. In particular, the significantly increased size of the nanodiscs can be useful in contexts where membranes are studied and as illustrated by the authors, for example, for the binding of peripheral membrane proteins. The approach presented in this work therefore has the potential to advance the field.

Response: We are gratified that the referee found our study has the potential to advance the field. We hope that the reviewer finds the revised manuscript satisfactory.

1. *Some important details, such as the detailed protein sequences, are missing and would prevent others from reproducing the work independently. These would need to be included to consider publishing the paper.*

Response: We have included all the protein sequences in the new Supplementary Table I and deposited all the plasmids in Addgene. In the revised manuscript, we have made the following changes:

“spMSP protein sequences are described in Supplementary Table I. All the spMSP plasmids have been deposited in Addgene: spMSP1D1 (ID: 173482), spNW15 (ID: 173483), spNW25 (ID: 173484), spNW30 (ID: 173485), spNW50 (ID: 173486), spNW80 (ID: 173487), spNW100 (ID: 173488)”.

2. *The authors cite and explain previously published systems that utilize the enzyme sortase for the cyclization of membrane scaffold proteins. In their work, the authors use the SpyCatcher/SpyTag system to enable this cyclization, however, they do not cite any of the previous work that used this type of cyclization, which has been used for other proteins.*

For example

Schoene, C., Fierer, J.O., Bennett, S.P. and Howarth, M. (2014), *SpyTag/SpyCatcher Cyclization Confers Resilience to Boiling on a Mesophilic Enzyme*. *Angew. Chem. Int. Ed.*, 53: 6101-6104. <https://doi.org/10.1002/anie.201402519>

Wang, J., Wang, Y., Wang, X., Zhang, D., Wu, S., & Zhang, G. (2016). *Enhanced thermal stability of lichenase from Bacillus subtilis 168 by SpyTag/SpyCatcher-mediated spontaneous cyclization*. *Biotechnology for biofuels*, 9, 79. <https://doi.org/10.1186/s13068-016-0490-5>

Si M, Xu Q, Jiang L, Huang H (2016) *SpyTag/SpyCatcher Cyclization Enhances the Thermostability of Firefly Luciferase*. *PLoS ONE* 11(9): e0162318. <https://doi.org/10.1371/journal.pone.0162318>

Schoene, C., Bennett, S. P., & Howarth, M. (2016). *SpyRing interrogation: analyzing how enzyme resilience can be achieved with phytase and distinct cyclization chemistries*. *Scientific reports*, 6, 21151. <https://doi.org/10.1038/srep21151>

Sun XB, Cao JW, Wang JK, Lin HZ, Gao DY, Qian GY, Park YD, Chen ZF, Wang Q. *SpyTag/SpyCatcher molecular cyclization confers protein stability and resilience to aggregation*. *N Biotechnol*. 2019 Mar 25;49:28-36. doi: 10.1016/j.nbt.2018.12.003.

These papers contain useful information for a discussion on the effects observed. Especially the paper from Sun et al also reports resilience to aggregation, which is similar to the decreased tendency of protein constructs of this work to be not soluble.

Response: In the revised manuscript, we have now cited and discussed these papers as the following:

“Recently, the Howarth group developed the SpyCatcher-SpyTag technology for protein ligation¹⁴. This new method exhibits outstanding kinetics and has enabled circularization of several proteins with enhanced stabilities¹⁵⁻¹⁹”.

3. *Some details are rather difficult to understand or are missing information to understand. For example,- Two non-circularized variants are mentioned (EQ and DA). I assume those are E -> Q and D-> A mutants? At which position in the protein? There are no citations to earlier works, which explain these mutants.*

Response: In the original manuscript, we followed the same nomenclature by the Howarth group on the EQ and DA mutants. And, it is correct that they stand for the E->Q and D->A mutation. In the revised manuscript, we have further explained these mutants, cited the proper reference, and made the following changes:

“To test if spMSP1D1 is circularized, we introduced two mutations that could abolish the covalent bond formation between SpyCatcher and SpyTag²². One mutant (EQ) is the substitution of Glu⁷⁷ to Gln in SpyCatcher, and the other (DA) is the substitution of Asp¹¹⁷ to Ala in SpyTag.”

4. - Citations for the specific MSP used as well as for the NW30 and NW50 protein are missing in the main text.

Response: MSP1D1 is from Bayburt et al., 2002 [ref. 6], whereas NW30 and NW50 are from Nasr et al., 2017 [ref.11]. We have made the following changes in the revised manuscript:

“To do so, we fused the SpyCatcher and SpyTag to the N- and C-termini of MSP1D1^{6, 7}, termed spMSP1D1 (Fig. 1A and B).”

“The success of spMSP1D1 prompted us to apply the same strategy onto large MSPs, known as NW30 and NW50¹¹, which could generate 15 and 50 nm nanodiscs via the sortase-mediated ligation approach, but suffers from low yields due to escalated aggregation and degradation elicited by their increased hydrophobic face.”

5. - Figure 2 needs more explanation. Information on the standard set of proteins mentioned can not be found in the figure legend nor the methods part. This information is relevant to understand the apparent difference in the size of the proteins between Figure 1B (SDS-PAGE) and Figure 2 top row (proteins with no lipid). The authors should comment on these apparent differences.

Response: We agree that the MWs of circularized MSP on SEC (Fig. 2) and SDS-PAGE (Fig. 1B) were very different. This is probably because circularized MSPs have a hollow ring structure, and their elution from SEC could not be directly compared to protein standards that are all globular proteins. In the revised manuscript, we have listed the protein standards used in SEC in the figure legends and made the following changes:

“We noted that the sizes of the proteins on SEC were much bigger than SDS-PAGE. This is probably because the circularized MSPs have a hollow ring structure, so their MWs could not be accurately determined by SEC calibrated using globular protein markers.”

“The proteins used for the calibration of the Superose 6 column are: thyroglobulin, 669kDa; ferritin, 443 kDa; β -amylase, 200 kDa; bovine serum albumin, 66 kDa.”

6. - Rhodamine-PE, the abbreviation PE is not explained.

Response: In the revised manuscript, we have described Rhodamine-PE in the main text and methods. We made the following changes:

“To validate these calculated values, we employed PC lipids doped with 0.5% N-(lissamine rhodamine B sulfonyl)-1,2-dipalmitoyl-sn-glycero-3-phosphoethanolamine (rhodamine-PE)”

“1,2-dioleoyl-sn-glycero-3-phosphocholine (PC), 1,2-dioleoyl-sn-glycero-3-phospho-l-serine (PS), 1-palmitoyl-2-oleoyl-sn-glycero-3-phosphoethanolamine (PE), 1,2-dioleoyl-sn-glycero-3-phospho-(1'-rac-glycerol) (PG), 1,2-dipalmitoyl-sn-glycero-3-phospho-ethanolamine-N-(7-nitro-2-1,3-benzoxadiazol-4-yl) (NBD-PE) and N-(lissamine rhodamine B sulfonyl)-1,2-dipalmitoyl-sn-glycero-3-phosphoethanolamine (rhodamine-PE) were obtained from Avanti Polar Lipids”.

7. - Details on the cloning are missing. It is well accepted upon fusion of proteins with tags such as the SpyCatcher/SpyTag, linker sequences can be important for protein function. Providing the full protein sequence for all fusion proteins would solve this problem.

Response: We do not have additional linker sequences in our constructs. We think that our constructs worked well because MSPs have a quite simple ring structure. In the revised manuscript, we have included all the protein sequences in the new Supplementary Table I and deposit all the spMSP plasmids at Addgene. Together, we have made the following changes:

“spMSP protein sequences are described in Supplementary Table I. All the spMSP plasmids have been deposited in Addgene: spMSP1D1 (ID: 173482), spNW15 (ID: 173483), spNW25 (ID: 173484), spNW30 (ID: 173485), spNW50 (ID: 173486), spNW80 (ID: 173487), spNW100 (ID: 173488)”.

8. - For figures S1, S2 and S4 error bars are depicted, but no info on the number of samples is given.

Response: We apologized for these mistakes. In the revised manuscript, we have now provided the n values in the figure legends and have moved the previous Fig. S4 into the new Fig. 5 as suggested by the second reviewer.

A technical question: in Figure S2 the protein content of the cNDs is calculated via their absorption at 280 nm. The lipids per cND are determined via the Rhodamine fluorescence. Based on 0.5% Rhodamine-PE doping roughly 1, 3 and 9 molecules per cND would result from the numbers presented, while there are two protein molecules in the cND. Since Rhodamine also has an absorption at 280 (please provide a spectrum/molecular extinction coefficient), how does that relate to the absorption of the proteins? Has this absorption been subtracted?

Response: In the original manuscript, we did not subtract the absorption of Rhodamine-PE, and we also could not find its extinction coefficient at 280 nm. To solve this problem, we now determined the spMSP protein concentrations using densitometry and recalculated the data in the new Fig. S3. In the revised manuscript, we have made the following changes in the figure legends:

“The lipid/protein ratios were calculated by determining protein concentrations via densitometry and lipid concentrations via the fluorescence emission of Rhodamine-PE.”

Reviewers' Comments:

Reviewer #1:

Remarks to the Author:

The authors have nicely revised the manuscript and addressed my concerns. I recommend it for publication.

Signed,

Michael Marty
Associate Professor
University of Arizona

Reviewer #2:

Remarks to the Author:

The authors have addressed most of my concerns. However, there are still issues with how they cite some of the work in the field. Examples are given below.

1) top of p7:

"For example, using 6 and 13 nm nanodiscs, previous studies were able to isolate nascent fusion pores (25), a crucial intermediate state formed during vesicle exocytosis."

Similar studies were done by others (refs. 27, 28) using small discs (~15-16 nm diameter) and should be cited here.

2) later in the same paragraph:

"Liposome-based reconstitution approach could study pore dilation (26), but it is difficult to parse out nascent fusion pores from dilated ones (27-30)."

Refs. 27-30 used nanodiscs, not liposomes.

3) 2nd paragraph on p.7:

"In line with previous studies [refs missing], the fusion pore was trapped in the initial open state using small spMSP1D1 nanodiscs, as a large cargo, 40 kDa fluorescein dextran, could not be released."

The previous studies are refs. 29 and 30.

4) top of p. 8:

"suggesting that the cooperative binding and oligomer formation of syt1 occurred only on the expanded lipid surface (34)."

It is good that ref. 34 is cited, but work on oligomerization of Syt1 goes much further back and some work by Ed Chapman and others should be cited here as well.

At the very least this is poor scholarship. Either the authors don't know the literature well, or worse, there is a deliberate attempt not to give due credit to others in the field.

Reviewer #3:

Remarks to the Author:

All my points have been addressed, thank you.

Reviewer #2 (Remarks to the Author):

The authors have addressed most of my concerns. However, there are still issues with how they cite some of the work in the field. Examples are given below.

1) top of p7:

"For example, using 6 and 13 nm nanodiscs, previous studies were able to isolate nascent fusion pores (25), a crucial intermediate state formed during vesicle exocytosis."

Similar studies were done by others (refs. 27, 28) using small discs (~15-16 nm diameter) and should be cited here.

2) later in the same paragraph:

"Liposome-based reconstitution approach could study pore dilation (26), but it is difficult to parse out nascent fusion pores from dilated ones (27-30)."

Refs. 27-30 used nanodiscs, not liposomes.

3) 2nd paragraph on p.7:

"In line with previous studies [refs missing], the fusion pore was trapped in the initial open state using small spMSP1D1 nanodiscs, as a large cargo, 40 kDa fluorescein dextran, could not be released."

The previous studies are refs. 29 and 30.

4) top of p. 8:

"suggesting that the cooperative binding and oligomer formation of syt1 occurred only on the expanded lipid surface (34)."

It is good that ref. 34 is cited, but work on oligomerization of Syt1 goes much further back and some work by Ed Chapman and others should be cited here as well.

At the very least this is poor scholarship. Either the authors don't know the literature well, or worse, there is a deliberate attempt not to give due credit to others in the field.

Response: We have no intention to deliberately ignore the contributions from others in the field, and we apologize for these mistakes in the literature. In the re-revised manuscript, we have corrected them accordingly and have made the following changes:

"For example, using 6 and 13 nm nanodiscs, previous studies were able to isolate nascent fusion pores²⁵⁻²⁷, a crucial intermediate state formed during vesicle exocytosis."

"Liposome-based reconstitution approach could study pore dilation²⁸, but it is difficult to parse out nascent fusion pores from dilated ones."

"In line with previous studies^{29, 30}, the fusion pore was trapped in the initial open state using small spMSP1D1 nanodiscs, as a large cargo, 40 kDa fluorescein dextran, could not be released."

"suggesting that the cooperative binding and oligomer formation of syt1 occurred only on the expanded lipid surface³⁴⁻³⁶."

We hope that the reviewer finds the revised manuscript satisfactory.